# Joint Sparsity for TomoSAR Imaging in Urban Areas Using Building POI and TerraSAR-X Staring Spotlight Data

**DOI:** 10.3390/s21206888

**Published:** 2021-10-17

**Authors:** Lei Pang, Yanfeng Gai, Tian Zhang

**Affiliations:** 1School of Geomatics and Urban Spatial Informatics, Beijing University of Civil Engineering and Architecture, Beijing 102616, China; panglei@bucea.edu.cn; 2School of Geosciences and Surveying Engineering, China University of Mining and Technology-Beijing, Beijing 100083, China; zhangtian@student.cumtb.edu.cn

**Keywords:** tomographic SAR, compressive sensing (CS), building POI, joint sparsity, TerraSAR-X staring spotlight

## Abstract

Synthetic aperture radar (SAR) tomography (TomoSAR) can obtain 3D imaging models of observed urban areas and can also discriminate different scatters in an azimuth–range pixel unit. Recently, compressive sensing (CS) has been applied to TomoSAR imaging with the use of very-high-resolution (VHR) SAR images delivered by modern SAR systems, such as TerraSAR-X and TanDEM-X. Compared with the traditional Fourier transform and spectrum estimation methods, using sparse information for TomoSAR imaging can obtain super-resolution power and robustness and is only minorly impacted by the sidelobe effect. However, due to the tight control of SAR satellite orbit, the number of acquisitions is usually too low to form a synthetic aperture in the elevation direction, and the baseline distribution of acquisitions is also uneven. In addition, artificial outliers may easily be generated in later TomoSAR processing, leading to a poor mapping product. Focusing on these problems, by synthesizing the opinions of various experts and scholarly works, this paper briefly reviews the research status of sparse TomoSAR imaging. Then, a joint sparse imaging algorithm, based on the building points of interest (POIs) and maximum likelihood estimation, is proposed to reduce the number of acquisitions required and reject the scatterer outliers. Moreover, we adopted the proposed novel workflow in the TerraSAR-X datasets in staring spotlight (ST) work mode. The experiments on simulation data and TerraSAR-X data stacks not only indicated the effectiveness of the proposed approach, but also proved the great potential of producing a high-precision dense point cloud from staring spotlight (ST) data.

## 1. Introduction

Synthetic aperture radar (SAR) has played an increasingly important role in earth observation and geological disaster monitoring due to the advantages of all-weather observations and short wavelengths (typically 3–25 cm). With the release of high-resolution spaceborne SAR satellites, such as TerraSAR-X and TanDEM-X, the efficient use of very high-resolution SAR images has become the focus of many scholars in urban mapping. In particular, as the new generation of SAR satellites can be controlled and extended to the staring spotlight (ST) observation mode, the image resolution has been improved even more, from 1.1 to 0.25 m [1,2,3,4]; however, because of the side looking imaging model of SAR system observations, it cannot distinguish different scatterers in the same azimuth–range pixel unit. To solve the problem of overlap in SAR images, the 3D imaging technology tomography SAR has been of great concern for researchers. Many scholars have also proposed many algorithms to be applied to TomoSAR imaging, such as Fourier transform methods and spectral estimation methods [5,6,7,8]. However, due to the limitation of the acquisitions and the uneven baselines of the spaceborne SAR system, the 3D reflectivity profile reconstructed by the algorithms above is usually terrible.

Considering that the ground observation scenes usually have strong sparsity and high anisotropy in the 3D field, the compressive sensing (CS) theory, proposed by Donoho in 2006, has been introduced into TomoSAR imaging. The CS theory holds that if a signal is sparse, it can be reconstructed by sampling points that are far below the requirements of the sampling theorem. This overcomes the limitation of the traditional Nyquist sampling theorem and proves the possibility of reconstructing the sparse signal [9]. In 2007, Baraniuk et al. first introduced the theory for the 3D imaging of TomoSAR and experimentally showed that the method could achieve focus in the elevation direction and that it had better imaging results [10]. Subsequently, many scholars have conducted extensive research based on CS theory for TomoSAR imaging and high-dimensional TomoSAR imaging [11,12,13,14,15,16,17,18,19,20,21,22]. In 2009, through simulation experiments, Budillion. et al. verified that the CS algorithm can effectively reduce the number of repeat orbit observations required for 3D reconstruction, and the subsequent experiments based on ERS satellite data successfully obtained the 3D image of the building [19]; on this basis, she then proposed many algorithms based on CS, such as ‘GLRT’ and ‘Fast-Sup-GLRT’, and carried out many experiments with TerraSAR-X/COSMO-SkyMed/Sentinel-1, and all the experiments achieved very good results; the details can be seen in [23,24]. Another outstanding team working on sparse TomoSAR imaging is Zhu. et al. from the Deutsches Zentrum für Luft-und Raumfahrt (DLR), which has made abundant achievements since 2009; for example, in [25] Zhu. X.X. et al. came up with the ‘SL1MMER’ algorithm, and her experiments proved that the algorithm has the advantage of super-resolution power and the elevation accuracy approximates the Cramér–Rao lower bound (CRLB). In 2014, Wang. et al. integrated a periodogram, SVD-Wiener, and ‘SL1MMER’ and proposed a novel workflow in sparse TomoSAR imaging; the test in downtown Las Vegas and the whole city of Berlin obtained reliable results. In addition to the algorithms described above, other scholars have also proposed many excellent algorithms, such as ‘Nonlocal CS-based’ [26], ’RIAA’ [27], and ‘TWIST’ [28]. On the whole, many experimental studies have confirmed that compared with traditional TomoSAR imaging methods, the CS-based algorithm has the advantages of high resolution, greatly reducing sampling data, and better reconstruction performance.

Although the TomoSAR imaging based on the CS algorithm has excellent performance in many aspects such as super-resolution power and sidelobe suppression [11,29], for a credible 3D reconstruction, the number of images required is usually very high (typically 40–60 scenes), and outliers are generated easily [30,31]. The huge cost undoubtedly restricts the application of TomoSAR imaging. Recently, with the development of big SAR data and in view of this problem, many scholars have worked towards using multi-source data or new SAR image data, such as polarization information, 3D semantic information in 2D images, high-precision DEM, TerraSAR-X ST data, and UAVSAR data, to assist TomoSAR imaging in reducing the required acquisitions and to provide more stable and reliable reconstruction results [30,32,33,34,35,36]. In [37], the authors investigated the application of polarization information to urban mapping and applied the ‘Fast-Sup-GLRT’ algorithm to dual polarization (HH + VV) data. Their experiment based on TSX data showed that dual-polarization data can outperform the single-polarization case and maintain a lower number of baselines. In [38], Wang et al. transformed the target signal reconstruction problem, with structural characteristics, into a Block Compressive Sensing (BCS) problem on the basis of the CS method, and $L_{1}/L_{2}$ norm optimization was performed for a target with sparse blocks. The experiments, based on RADARSAT-2 data, verified the effectiveness of this method. In the M-SL1MMER method, described in [31], 3D reconstruction of the observation was attained using GIS 2D footprints (positioning accuracies varying within 4 m); the drawback of this method is the high computational cost caused by the compensation of any orientation and/or shifting inaccuracies. Even so, experiments based on only six interferograms of TSX/TDX data demonstrated the effectiveness of the method in reducing the number of images required. Through the above summary and analysis, it is feasible to introduce multivariate information into TomoSAR imaging, and this has a much broader research scope for scholars.

In this paper, we propose an improved joint sparse TomoSAR imaging algorithm, which is ‘object-based’, intending to reduce the number of acquisitions needed. Compared with the conventional compressive sensing (CS) algorithm (e.g., OMP), the improved algorithm performs more accurately and is reliable both in building reconstruction and scatterer estimation. Note that ‘object-based’, here, means fully exploiting the prior information of the individual building (e.g., building POIs) to support TomoSAR imaging [1]. To demonstrate its effectiveness, the proposed workflow was applied to TerraSAR-X datasets in the staring spotlight (ST) work mode and we carried out a comparative analysis for the four aspects of elevation estimation accuracy, super-resolution power, detection rate, as well as the great potential to produce a high-precision dense point clouds. It should be added that the ST datasets used in this study were Very High Resolution (VHR) data (0.25 m); more details about the test area and ST basic can be found in Section 5 as well as in [39,40]. In Section 2, the principles of sparse TomoSAR imaging are presented, including the TomoSAR imaging model and the compressive sensing theory. In Section 3, we summarize the concept of joint sparsity and introduce the procedure of extracting the building Line-of-Interest (LOI), building mask, and iso-height lines from building POIs. In Section 4, we verify the proposed workflow in three aspects—elevation estimation accuracy, super-resolution capability, and detection rate—through simulation data. In Section 5, the imaging experiments are described, which were carried out using ST mode datasets of TerraSAR-X in Shenzhen, China. We conclude in Section 6 with a discussion and suggestions for future work.

## 2. TomoSAR Imaging Based on Compressive Sensing

### 2.1. TomoSAR System Mode

TomoSAR is developed from InSAR technology, borrowing some principles from medical CT imaging technology. As InSAR can only obtain the elevation of the observed area, with the assumption that there is only one dominant scatter in one azimuth–range pixel. TomoSAR technology constructs the synthetic aperture in the elevation direction, which is perpendicular to the plane formed by the range (r) and azimuth (x), and it can solve the problem of several dominant scatterers located at different heights, but projected in the same azimuth–range resolution cell. Considering the huge cost of constructing a multi-antenna in the elevation direction, TomoSAR technology obtains the ground image data through repeated observations at different heights and times in the same area [7,41,42], as shown in Figure 1.

In Figure 1, x is the azimuth direction, r is the range direction, and s is the elevation direction. After registration and phase correction are performed, the focused complex-valued measurement $g_{n}$ of one azimuth–range pixel for the *n*th acquisition at aperture position $b_{n}\;$ can be obtained. Each pixel value can be expressed as the integral of the backscattering distribution along the elevation [7], which can be expressed as:
(1)$$g_{n}=\overset{}{\underset{∆s}{\int }}\gamma \left(s\right)\exp\left({-j2\pi \xi }_{n}s\right)\mathrm{ds},\;n=1,2,3,\ldots ,N\;.$$
where γ(s) is the back reflectivity function along the elevation direction of the imaging area, the spatial sampling (elevation) frequency $\xi_{n}$ can be calculated as $\xi_{n}{=-2b}_{⊥n}/(\lambda r)$, $b_{⊥n}\;$ is the vertical baseline distance, λ is the wavelength, ∆s is the elevation extent that depends on the antenna diffraction pattern width, and r is the central range. After discretization, Equation (1) can be approximated, simply, as:(2) g=R γ+ε,
where g is the measurement vector of length N, R is the dictionary matrix with the size of N×L, $R_{\mathrm{nl}}=\exp(-j2\pi \xi_{n}s_{l})$, and γ is γ(s) irregular sampling of elevation at every position (1,..., L), and ∆b is the aperture size. Similar to the azimuth resolution, the Rayleigh resolution is inversely proportional to the aperture size, $\rho_{s}=\lambda r/2∆b$. As L >> N, the system model (2) is severely ill-posed [15].

### 2.2. Compressive Sensing

The core idea of compressive sensing theory is the reduction of the dimensionality of a high-dimensional sparse matrix while, at the same time, preserving the signal quality, in order to conduct signal sampling and compression simultaneously. The compressive sensing theory allows us to exceed the Shannon limit by using the sparse characteristics of the signal, which constitutes a significant step in the signal process. The mathematical model of the reconstruction of compressive sensing is as follows:(3) g=Φ Χ,
assuming that the original signal is Χ=Ψγ, with a length of L, where Ψ is the sparse orthogonal basis matrix of the original signal Χ. If the original signal has k non-zero values, it can be called k-sparse. In Equation (3), g is the measurement matrix, where $g\in R_{n}$(n = 1,2,3…N), and where Φ (with size N × L) is the sensing matrix (also called the dictionary matrix), which corresponds to the sub-sampling process and projects a high-dimensional signal x into a low-dimensional space.

The problem of compressive sensing is to obtain the original signal Χ by solving the underdetermined equations on the basis of a known measurement value g and sensing matrix Φ; however, under normal circumstances, the number of equations is far less than the number of unknowns, so the equation has no definite solution, and the signal cannot be reconstructed. As the signal is k-sparse (k non-zero numbers), if the Φ in the above formula satisfies the restricted isometry property (RIP) [43,44,45], then k can be accurately reconstructed from N measured values. Equation (3) can be seen as satisfying the following equation (ignoring the noise):(4)$$\underset{\gamma }{\min}{\left\|\gamma \right\|}_{0}s.t.\;g=\Phi \;Χ.$$

According to CS theory, if the sensing matrix R fulfills the incoherence properties, minimization of the convex $L_{1}/L_{2}$ norm provides the same solution as the NP-hard $L_{0}$ norm minimization. Thus, Equation (4) can be approximated as:(5)$$\;\underset{\gamma }{\;\min}{\left\|\gamma \right\|}_{1}s.t.g=\Phi \;Χ.$$

There are many solutions for Equation (5), such as convex optimization, greedy pursuit, and the Bayes algorithm, which are not discussed here; detailed information can be found in [14]. The application of the compressive sensing theory to TomoSAR imaging mainly includes the following four steps: (1) sparse representation of the SAR echo signal; (2) the construction of the mapping matrix, which is the key step to realizing sparse TomoSAR imaging; (3) the tomographic focus reconstruction algorithm, which is the core step of TomoSAR imaging and directly determines the performance of imaging; and (4) parameter estimation and model evaluation [25,46,47,48]. Note that, in dense urban areas, excepting the volumetric scatterers with the character of temporarily decorrelating, there are typically only 0–4 dominant scatterers in the elevation of one Az–Rg pixel, which can be considered sparse in the space domain of elevation.

## 3. Materials and Methods

In this section, the principle of joint sparsity based on building prior knowledge (POIs) and maximum likelihood estimation to reduce the number of acquisitions needed and estimate scatterers is introduced First, the relevant concepts and principles of joint sparsity are summarized, and the extraction of the building LOI, building mask, and iso-height lines from building POIs is discussed in detail. The maximum likelihood estimation model for scatterers estimation based on the Bayesian information criterion (BIC-MLE) was subsequently applied to remove the outliers [49]. The overall technical route is shown in Figure 2 below; the preprocessing of the input single-look slant-range complex (SSC) images was performed by a persistent scatterer interferometry (PSI) process, as well as deramp and so on [50].

### 3.1. Joint Sparsity Basic

Joint sparsity has been developed on the basis of compressive sensing (CS) for sparse reconstruction [51,52]. In the theory of joint sparsity for sparsity reconstruction, the prior structural information in the observed scene is taken into full consideration. The joint sparsity for sparsity reconstruction is realized by incorporating sparsity constraints and structural constraints into the reconstruction imaging processing, which can contribute to a new sparsity scene according to the prior structural information that is excavated. Joint sparsity not only emphasizes the sparsity characteristics in traditional CS theory, but also improves its disadvantages of being greatly affected by noise and having pseudo-values [31,53]. Its principle is also derived from Equation (2):(6)$$G_{N\times M}{=\psi }_{N\times L\;}\Upsilon_{L\times M\;}{+E}_{N\times M}.\;$$

In this study, the building LOI, building masks, and iso-height lines were used as prior information to construct a new sparse scene; furthermore, the reconstruction of the individual building was realized using the structural characteristics of the pixels on the same iso-height line, which share the same spatial spectrum and elevation simultaneously. Suppose that in one iso-height line there are a total of M pixels. Then, we can infer that every pixel’s $\xi_{n}$ is equivalent, which can be defined for each pixel’s dictionary matrix $R_{1}$ ≈ $R_{2}$ … ≈ $R_{M}$. The expressions of each element in Equation (6) can be obtained as follows:$$G=\left[\;\begin{array}{ccc} \begin{array}{c} g_{1\times 1}\\ \begin{array}{c} g_{2\times 1}\\ \begin{array}{c} \vdots \\ g_{N\times 1} \end{array} \end{array} \end{array} & \begin{array}{c} \begin{array}{c} g_{1\times 2}\\ g_{2\times 2} \end{array}\\ \begin{array}{c} \vdots \\ g_{N\times 2} \end{array} \end{array} & \begin{array}{c} \begin{array}{c} \begin{array}{cc} \cdots & g_{1\times M} \end{array}\\ \begin{array}{cc} \cdots & g_{2\times M} \end{array} \end{array}\\ \begin{array}{cc} & \vdots \end{array}\\ \begin{array}{cc} \cdots & g_{N\times M} \end{array} \end{array} \end{array}\right]\;E=\left[\;\begin{array}{ccc} \begin{array}{c} \varepsilon_{1\times 1}\\ \begin{array}{c} \varepsilon_{2\times 1}\\ \begin{array}{c} \vdots \\ \varepsilon_{N\times 1} \end{array} \end{array} \end{array} & \begin{array}{c} \begin{array}{c} \varepsilon_{1\times 2}\\ \varepsilon_{2\times 2} \end{array}\\ \begin{array}{c} \vdots \\ \varepsilon_{N\times 2} \end{array} \end{array} & \begin{array}{c} \begin{array}{c} \begin{array}{cc} \cdots & \varepsilon_{1\times M} \end{array}\\ \begin{array}{cc} \cdots & \varepsilon_{2\times M} \end{array} \end{array}\\ \begin{array}{cc} & \vdots \end{array}\\ \begin{array}{cc} \cdots & \varepsilon_{N\times M} \end{array} \end{array} \end{array}\right]$$
where $\psi_{N\times L}=\left[\;\begin{array}{ccc} \begin{array}{c} {\exp(-j2\pi \xi }_{1}s_{1})\\ \begin{array}{c} {\exp(-j2\pi \xi }_{2}s_{1})\\ \begin{array}{c} \vdots \\ {\exp(-j2\pi \xi }_{N}s_{1}) \end{array} \end{array} \end{array} & \begin{array}{c} \begin{array}{c} {\exp(-j2\pi \xi }_{1}s_{2})\\ {\exp(-j2\pi \xi }_{2}s_{2}) \end{array}\\ \begin{array}{c} \vdots \\ {\exp(-j2\pi \xi }_{N}s_{2}) \end{array} \end{array} & \begin{array}{c} \begin{array}{c} \begin{array}{cc} \cdots & {\exp(-j2\pi \xi }_{1}s_{L}) \end{array}\\ \begin{array}{cc} \cdots & {\exp(-j2\pi \xi }_{2}s_{L}) \end{array} \end{array}\\ \;\vdots \\ \begin{array}{cc} \cdots & {\exp(-j2\pi \xi }_{N}s_{L}) \end{array} \end{array} \end{array}\right];$

Similarly, the optimum estimate $\Upsilon_{L\times M\;}$ with CS in Equation (6) is found to be:(7)Υ ^=arg minΥ{12‖G−ψΥ‖+22λκ‖Υ‖1,2}.

After solving (7) described in [54,55], the BIC-MLE model was utilized to estimate the optional number for each pixel individually. According to the above principles, the building’s prior information extraction process is as follows. Algorithm 1 is the corresponding pseudo-code of the process.


**Algorithm 1: Procedure to Extract LOI, Mask, and Iso-Height Lines from Building POI.**
1: #Generate the LOI;2: Import image and identify the POI of the test building by geodetic surveying;3: Connect the POI facing the SAR sensor side to form the LOI and transform it to SAR coordinate system by geocoding;4: Initialize the max-shift range of the surveyed area and find the pixels that LOI passed;5: #Generate Mask;6: While “range shift ≤ range limit”, one must:7:  Shift the LOI in the range direction by a distance of 1 and find the pixels passed in every shift;8:  Compute the pixel average intensity value of every shift;9:  Compute the intensity difference value between every pixel shift and the LOI;10: Find the maximum difference value;11: Otherwise, break;12: #Generate iso-height lines;13: While “range shift ≤ range limit of Mask”, one must;14: Shift the LOI only in the range direction by the sub-pixel distance;15: Compute the distance between a pixel and its adjacent iso-height lines and find the closest iso-height line to the pixel;16: The pixels belonging to the same iso-height line are associated;17: Construct new sparse scenario;18: Otherwise, break;19: # BIC-MLE;20: Initialize K == 0, while K == 0–4;21: Calculate each model based on the BIC for every pixel;22: Find the model that best fits each pixel, and estimate the elevation, amplitude, and phase;23: K = K + 1;24: Otherwise, end.

### 3.2. Building LOI

In the backscatter mechanism of an individual building in SAR illumination, due to the dihedral corner structure formed by the ground and the building wall, the echo energy in most cases is superimposed and converged, which results in SAR images often presenting with a very bright “L”-shaped strip. Meanwhile, due to the nature of the side-looking geometry, the SAR beam can only obtain one side of the building’s wall facade and the top surface of the building [42]; thus, we chose the building’s bottom “L”-shaped line facing the SAR sensor side as the LOI. In order to obtain the LOI of the individual building, we should first obtain the precise coordinates of the Points of Interest (POIs) facing the SAR sensor of the building, through the geodetic survey method. It is important to note that if a POI of the building facing the SAR line of the sight direction is determined by its projection on the range axis line (assuming that the line is range = 0), when the projection line of the POI has no intersection with the building contour line that exists, the POI will be regarded as facing the side of the SAR sensor. Secondly, we converted the LOI, formed by the POIs that form the world (latitude/longitude) coordinate, to the SAR imaging coordinate system (accuracy on the order of 1/4 pixel). The coordinates of the measured POIs and pixels of the LOI in the SAR coordinate system (after being geocoded) are shown in Table 1 and Figure 3 respectively:

### 3.3. Building Mask

After obtaining the LOI of the individual building, we subsequently obtained the building mask. Here, we defined the building LOI as the initial mask of the individual building, and the building mask can be acquired by shifting the initial mask. The principle of the method to obtain the building mask is to iterate the initial mask of the building only in the range direction, and the maximum shift distance is determined by the tallest building in the observation area. In the local area, the tallest building was about 225 m; hence, the maximum pixel amount in the SAR image was 225/0.588 ≈ 382 (0.588, here, is the range resolution). The shift step size was one pixel. By calculating the average intensity of all the pixels that the initial mask passes through in every iteration, the iteration with the greatest change in the average intensity will be found. Furthermore, this iteration was regarded as the last translation of the initial mask, and the contour obtained by this shift was precisely the building mask. The following figure shows the max-shift mask and the building mask overlaid on the intensity image.

### 3.4. Building ISO-Height Lines

The purpose of obtaining the building mask was to obtain the range limitation of the iso-height lines. The method used to obtain the iso-height lines was similar to that for the building mask; however, the initial mask was shifted with a sub-pixel step length, and the maximum shift range was the boundary of the building mask. After all the iso-height lines were accepted, all the pixels were preceded by pixel association. The meaning of pixel association is to join the pixels on the same iso-height line. As the iso-height line was shifted by the step length of the sub-pixel, an extra calculation was needed to calculate the distance between each pixel and the two nearest iso-height lines. It should be noted that whether the distance belongs to the nearest distance is determined using the magnitude of the fitting error. After the association of pixels, a new structured sparsity scene was constructed for sparsity inversion. Figure 4c, above, provides a schematic diagram of the iso-height lines.

## 4. Verification of Simulation Data

In this section, we verified the proposed joint sparsity imaging workflow through simulation data. The verification criterion mainly included three aspects: the scatterer elevation estimation accuracy, super-resolution ability, and scatterer detection rate. In order to verify the effectiveness of the imaging workflow proposed in this paper, we conducted a comparative analysis between the proposed workflow and the traditional TomoSAR imaging algorithm based on Orthogonal Matching Pursuit (OMP). Note that the OMP algorithm used in this paper was Stanford University’s sparse reconstruction toolkit, Sparselab (http://sparselab.stanford.edu/). In the process of pixel association on the above iso-height lines, the average number of pixels in each iso-height line was 29. According to this, we set M = 29 in the simulation data. Given that SNR = 10, the number of images was set as N = 10/18, and the number of scatterers was set as 0–2 per pixel. As the OMP for TomoSAR imaging proceeds pixel by pixel and does not require pixel associations, we combined the 29 pixels for the purpose of convenient drawing.

Figure 5 shows the elevation estimation accuracy when the number of images was different. The solid blue line represents the original signal; the solid red circle represents the signal recovered by the joint sparsity algorithm, and the black solid dot was recovered by OMP. Figure 5c shows the elevation estimation accuracy errors in different image numbers. In order to make the comparison more intuitive, we squared and summed the elevation estimation error of the scatterers in each pixel. It can be seen from the figure that as the number of images decreased to 10, the signal elevation estimation error recovered by the OMP algorithm increased, and there were also many noise points that appeared. The Table 2 below shows the Root Mean Square Error (RMSE) of elevation estimation accuracy reconstructed by different algorithms with different images. The curve of the cumulative deviation of N = 10/joint sparsity was much smaller than that with N = 18/OMP, which verified the effectiveness of the novel workflow.

Figure 6 shows a comparison of the super-resolution capability when the number of images dropped; the elevation distance (D) was set to 20, 10, and 5. Table 3 is the corresponding reflectivity mean value of the outliers (only scatterers whose elevation accuracy exceed ±5 can be considered as outliers). As can be seen in the picture, when N = 18, with the reduction in elevation distance, the super-resolution capability of OMP was gradually decreased, and the number of miscellaneous points generated also increased. In Figure 6c, the signal recovered by OMP had an obvious false value, and part of the scatterers could not be separated effectively. When N = 10, only a few scatterers could be recovered accurately by OMP, but with many false values generated as illustrated in Table 3. However, the joint sparsity imaging algorithm still had powerful separation adaptability to the scatterers with a small distance, even at N = 10, while, at the same time, maintaining a high elevation estimation accuracy.

Figure 7 is the scatterer detection rate under the influence of the acquisition number and elevation distance. Note that the scatterer detected is only included in the calculation when its elevation estimation accuracy reaches half of the set elevation distance. It can be seen from the figure that as the elevation distance of scatterers decreased, the detection rate of OMP decreased dramatically and only had a small increase between D = 0.2–0.5 m, regardless of whether N = 18 or N = 10. On the other hand, the joint sparsity imaging algorithm had greater adaptability to the number of images and could achieve a higher scatterer detection rate. When the distance exceeded about 0.3 m, the detection rate of joint sparsity was almost impervious to the number of acquisitions.

The above simulation experiments demonstrate that the proposed workflow had strong robustness in the face of a decrease in the number of images and a decrease in the scatterer interval. It showed a good reconstruction effect, regardless of the accuracy of the point location estimation, the super-resolution ability, or the scatterer detection rate, which fully verifies the effectiveness of the novel workflow. In the next section, we applied the workflow to real satellite data in order to present a more convincing verification.

## 5. Practical Test Results

This section presents a case where an individual building was retrieved using a stack of TerraSAR-X acquisitions in ST mode. The test area was a building of 400 × 450 pixels in size in Baoan, Shenzhen, with a time span of 792 days (from November 2015–January 2018). Its temporal and spatial baseline digraph can be seen in Figure 8. According to the geographic survey data, the test building had a total of 48 floors and was about 162 m tall. Furthermore, the Rayleigh resolution of the test area in elevation direction was about 21 m. The parameters of a ST mode image in the test area are shown in the Table 4 below:

Figure 9 and Figure 10 show the experimental results of the test area. The experimental results are the reconstruction results of the two algorithms and the point cloud of the separated single and double scatterers when N = 18/10, respectively. It can be seen from Figure 10 that when the number of images was reduced to 10, the reconstruction results of different algorithms had a large order of magnitude reduction in the density of point clouds, and there was also a large order of magnitude difference in the scatter miscellany points. When N = 18, a total of 2113 scatterers were detected by the joint sparsity algorithm, while 1352 scatterers were detected by OMP, and the common scatters (containing double scatterers with an average error less than 0.1 m) approached 970. Although there was a small difference in the number of scatterers detected by the two algorithms, it can be clearly seen from Figure 9b that there were few false scatterers in them. In addition, when N = 18, the maximum height recovered by the joint sparsity was 161.4 m, and that reconstructed by the OMP was 161.1 m, both of which were consistent with the actual height of the test building.

It can be seen from Figure 9a–d that when the number of images decreased, the number of scatterers detected by the two algorithms decreased by a large order of magnitude (N = 10/joint sparsity: 1015; N = 10/OMP: 1891; common scatter: 461). However, compared with the OMP, the reconstruction results recovered by joint sparsity were relatively robust, which strongly verifies the effectiveness of the method proposed in this paper. Moreover, when the number of acquisitions was reduced to 10, the number of scatterers detected by the joint sparsity method was not significantly different from that detected by the OMP method when N = 18, which fully demonstrates that the introduction of building prior information could effectively assist TomoSAR imaging in achieving a reasonable result.

Figure 10 shows the influence of the number of images and different algorithms on the ability to distinguish scatterers. As can be seen from Figure 10, when the number of images decreased, the number of double scatterers decreased to a great extent. For example, in Figure 10c, the number of double scatterers detected decreased by at least half, compared with that in Figure 10a (452 in total). In addition, the number of false scatterers increased greatly. One strong piece of evidence is that the number of double scatterers detected in Figure 10d was 2.6 times that in Figure 10b (502 in total). Significantly, however, the joint sparsity still had a strong ability to distinguish scatterers when the number of images decreased.

Finally, on the other hand, it is worth mentioning the different computational burdens of the two algorithms. Under the same conditions, the calculation time for the joint sparsity was about 40 times that of the OMP algorithm (joint sparsity: 22.6 h, OMP: 0.52 h). Of course, in addition to the difference in convergence speed between joint sparsity and OMP, the addition of an extra calculation and the penalty function term to select the number of scatterers is also an inactive factor.

## 6. Conclusions

In summary, in this paper, an ‘object-based’ compressive sensing imaging workflow was proposed in TomoSAR imaging to solve two problems: the large number of acquisitions needed and the outliers in scatterer estimation. On the basis of conventional compressive sensing and considering the sparsity in the observed area, this novel workflow makes full use of prior information about an individual building, such as building POIs, to construct new structural sparse scenes. Furthermore, a maximum likelihood function model based on the Bayesian information criterion for scatterer estimation was applied to reject outliers. The simulation results showed that the algorithm exhibited good performance in elevation estimation accuracy, super-resolution ability, and detection rate. In terms of the RMSE of elevation estimation accuracy, the joint sparsity was 1.097 (N = 10) compared with 0.919 (N = 18) of OMP. Regarding the reflectivity mean value (D = 05) of the outliers, the joint sparsity was 0.025 (N = 10) compared with 0.139 (N = 18) of OMP. In addition, experiments based on TerraSAR-X staring spotlight (ST) datasets in the Shenzhen area not only demonstrated the superiority of the workflow in SAR tomography, but also indicated the great potential of high-precision dense point cloud generation from ST stacks.

If we consider the limitation of the proposed algorithm, the huge time cost and prior information preparation is still somewhat of a problem. In terms of processing time of one pixel, the OMP was only 0.012 s, and the joint sparsity was 0.41 s; even without optimization, the joint sparsity still reached 0.29 s. In particular, the preparation of prior information required us to conduct field investigations and measurements, which also limits the application of the joint sparsity in emergency reconstructions and large-scale scene reconstructions to some extent. In order to promote the development of the algorithm in engineering applications, we intend to conduct in-depth research on the problem of the high computational cost of the algorithm as well as the optimization of prior information extraction. We will also extend the workflow to higher-dimensional imaging, such as differential TomoSAR imaging (D-TomoSAR) and 5D-SAR imaging in the future. Last but not least, the high-precision dense point clouds in which we have a great interest will also be conducted for further experimental analyses.

## Figures and Tables

**Figure 1 sensors-21-06888-f001:**
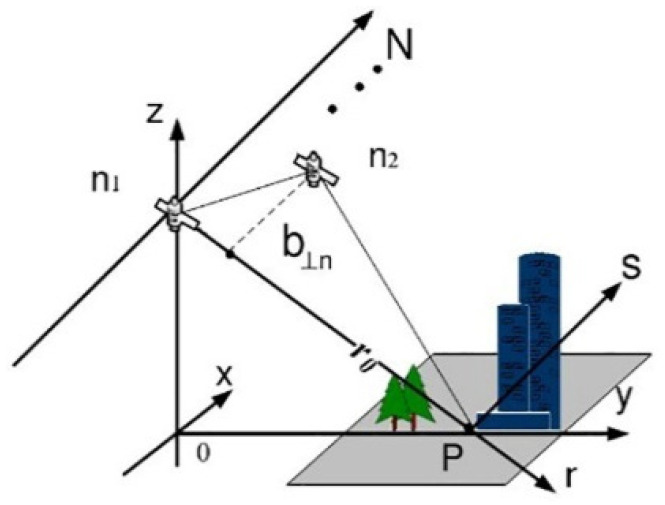
The imaging mode of TomoSAR.

**Figure 2 sensors-21-06888-f002:**
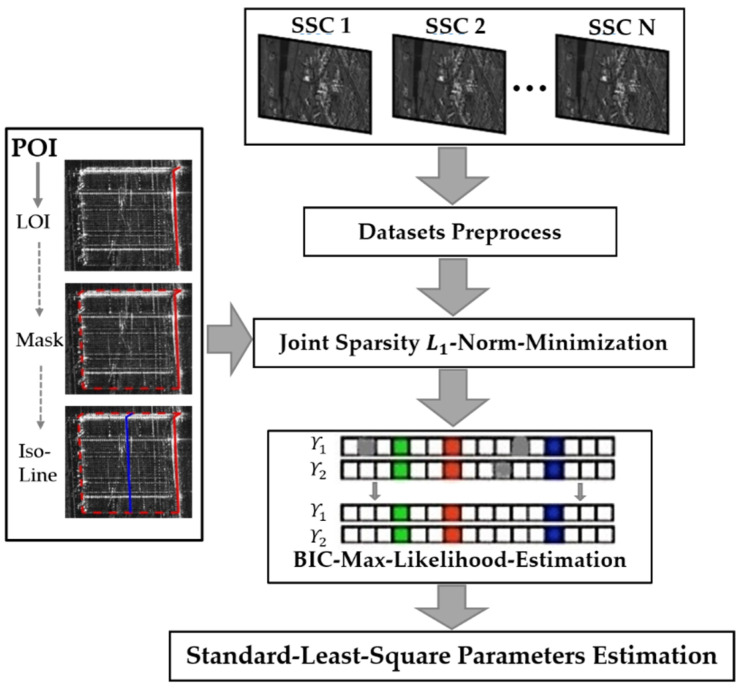
General workflow.

**Figure 3 sensors-21-06888-f003:**
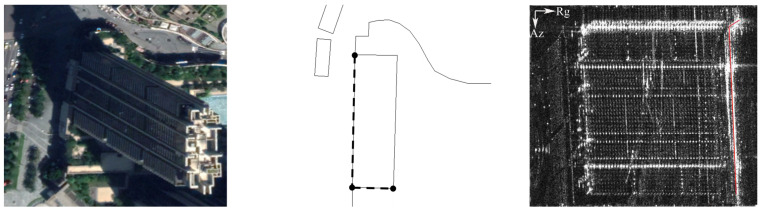
The precise coordinates of POI (WGS84).

**Figure 4 sensors-21-06888-f004:**
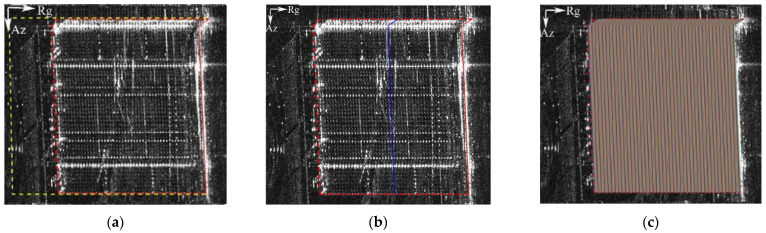
Mask and iso-height-lines overlaid on the intensity map: (**a**) max-shift mask (yellow dotted line) and building mask (red dotted line); (**b**) a stochastic iso-height line in blue; and (**c**) iso-height lines with random color.

**Figure 5 sensors-21-06888-f005:**
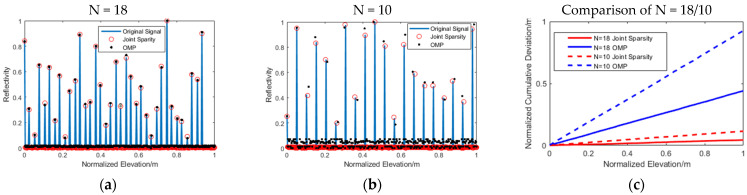
Elevation estimation accuracy map: (**a**) N = 18; (**b**) N = 10; and (**c**) cumulative deviation comparison of N = 18/10. The blue solid line represents the original signal; the red circle is the signal recovered by the joint sparsity algorithm, and the black dot is recovered by OMP.

**Figure 6 sensors-21-06888-f006:**
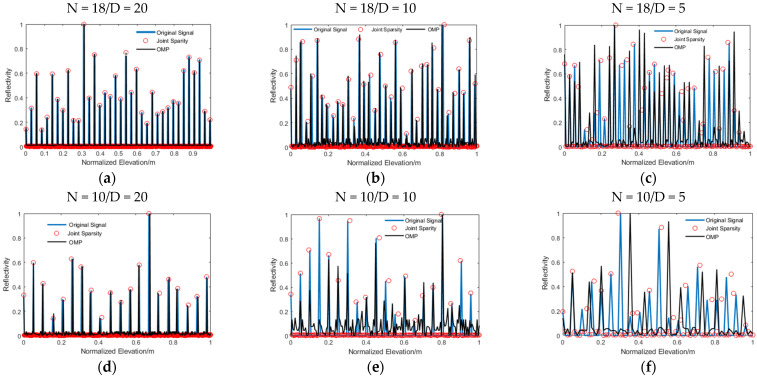
The super-resolution capability map: (**a**) N = 18/D = 20; (**b**) N = 18/D = 10; (**c**) N = 18/D = 5; (**d**) N = 10/D = 20; (**e**) N = 10/D = 10; and (**f**) N = 10/D = 5. The blue solid line represents the original signal; the red solid circle is the signal recovered by the joint sparsity algorithm; and the black solid line is recovered by OMP.

**Figure 7 sensors-21-06888-f007:**
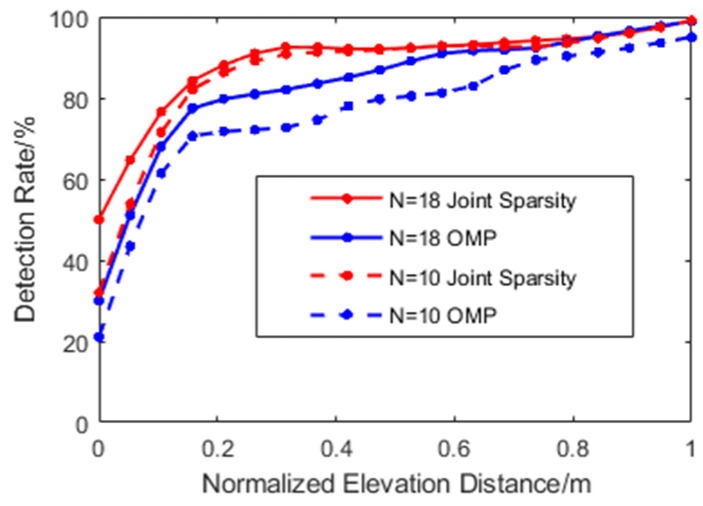
Comparation of the detection rate. Distance (D) is normalized from 0, 1–20.

**Figure 8 sensors-21-06888-f008:**
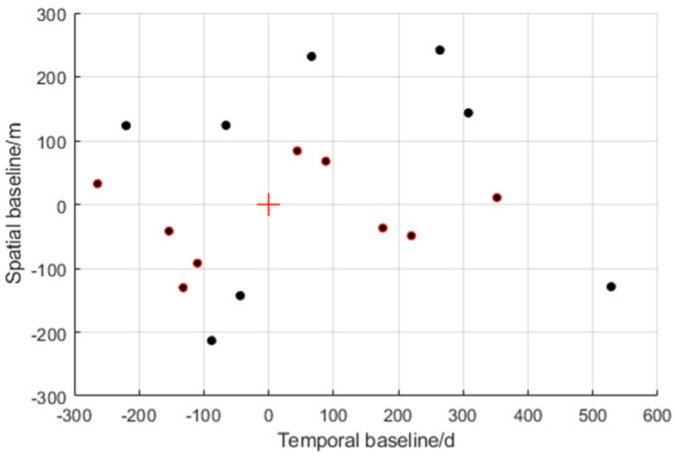
Spatial and temporal baselines of the TerraSAR–X data stack over Shenzhen. The dots with red edges represent the baseline data in the experiment N = 10.

**Figure 9 sensors-21-06888-f009:**
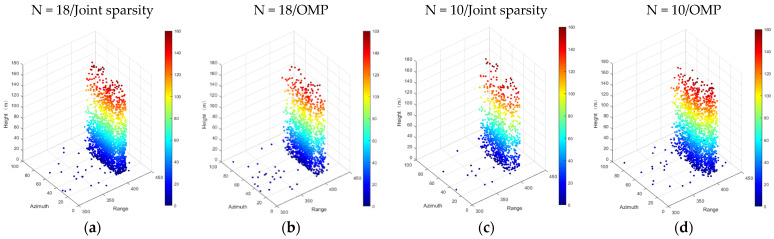
Color-coded reconstruction of the test individual building in Shenzhen. From left to right, (**a**) N = 18/joint sparsity, (**b**) N = 18/OMP, (**c**) N = 10/joint sparsity, and (**d**) N = 10/OMP.

**Figure 10 sensors-21-06888-f010:**
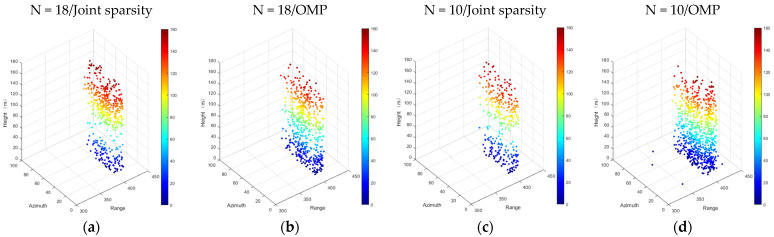
Unmixing of scatterers detected by different algorithms of the test individual building in Shenzhen. From left to right: (**a**) N = 18/joint sparsity, (**b**) N = 18/OMP, (**c**) N = 10/joint sparsity, and (**d**) N = 10/OMP.

**Table 1 sensors-21-06888-t001:** The precise coordinates of POI (WGS84).

POI	Latitude/Deg	Longitude/Deg	Height/M
A	22.55513197	113.88121665	0.987
B	22.55513975	113.88109875	1.012
C	22.55564238	113.88098455	0.995

**Table 2 sensors-21-06888-t002:** The RMSE of elevation estimation accuracy.

RMSE	Joint Sparsity	OMP
N = 18	0.554	0.919
N = 10	1.097	1.679

**Table 3 sensors-21-06888-t003:** The reflectivity mean value of the outliers. The two algorithms correspond to the two columns of numbers; the left column is N = 18 and the right column is N = 10.

Mean Value	Joint Sparsity	OMP
D = 20	0.012 0.017	0.018 0.024
D = 10	0.013 0.019	0.028 0.129
D = 05	0.022 0.025	0.139 0.279

**Table 4 sensors-21-06888-t004:** Some parameters of a TSX ST acquisition of Shenzhen.

Some Parameters of a TerraSAR-X Staring Spotlight Acquisition of Shenzhen	
Incident Angle	35.380°
Polarization Mode	HH
Number of Azimuth Beams	113
Azimuth Steering Angle	±2.210°
Azimuth Resolution	0.230 m
Slant Range Resolution	0.588 m
Scene Azimuth Extent	3052.988 m
Scene Range Extent	6262.264 m
Common PRF	42,400 Hz
Azimuth Look Bandwidth	38,292.780 Hz
Range Look Bandwidth	300 MHz
Scene Duration Time	0.43 s
RAW Duration Time	6.73 s

## Data Availability

Not applicable.

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
