# Peer review of "Joint Sparsity for TomoSAR Imaging in Urban Areas Using Building POI and TerraSAR-X Staring Spotlight Data"

_sensors, 2021, doi:10.3390/s21206888_

Round 1

Reviewer 1 Report

The manuscript discribes a object based compressive sensing workflow for tomographic imaging based on apriori building information.

My recommendation is to accept the manuscript, subject to major revision.

Major comments:

* Overall: it is hard to understand exactly what is review of existing theory
  and what is contribution of this manuscript. Please improve the emphasis on
  what is the main contribution.

* Page 6, all equations: It is hard to understand the dimensionality of the various matrices.

 - Why is the \psi matrix subscripted with "1 x M"? If I read this correctly,
   this matrix has the dimension (N x LM)?
 - What is the definition of the matrix \Upsilon_{L x M}", and its dimensionality? 
 - The definition of the submatrices \phi_{1 x m} on line 222 is very imprecise,
   and mixes uppercase and lowercase letters in a confusing way.

* Page 7, Figure 4: The figure is almost unreadable when printed on a normal
  office printer, and hardly readable on screen. In particular subfigure (4b),
  but also (4a) and (4c). Please improve the readability of the figure.

* Page 8, Figure 5b: This figure is unreadable. Please simplify, e.g. by
  plotting only a few isoheight lines, with a single color.

* Only one building is used in both the simulations and . I miss a thorough
  discussion on the generality of the method (including limitation), given that
  a very precise apriori information about the object is required.

* Page 13, line 414-419: A better analysis of the algorithmic complexity should
  be done. Just stating 40x higher processing time than a presumably well
  optimized software package (for OMP) is not very useful. How much of this is
  due to a more complicated algorithm and how much is due to non-optimized
  coding of the presented algorithm? What is the typical processing time for a
  single object/building? Seconds? Minutes? Hours?

* Discussion point: How is the algorithm expected to perform in case of lower, but still
  high-resolution SAR data (2-3 m), which is more realistically available?

* The discussion on sidelobe suppression must be improved. Why does the proposed
  methodology more robustly reject sidelobes, compared to other algorithms?

Minor comments and suggestions:

* Page 3, Eq 1: The domain of integration is denoted as "\Delta s", without any
  definition.

* Page 3, line 128: "\Delta b" is at best implicitly defined (as "the aperture
  size", which is imprecise).

* Page 3, line 125-127: how is the theoretical dimension L of the dictionary matrix calculated?

* Page 4, line 161-163: it is claimed that "[...] for the X-band high-resolution
  SAR satellite, there are only 1-3 dominant scatterers in the elevation of one
  Az-Rg pixel [...]". In the elevation direction, the number of scatterers in a
  resolution cells elevation dimension is not dependent on either X-band or
  high-resolution, but on the scattering environment. (However, these properties
  influences the number of scatterers in the azimuth-range dimensions.) In dense
  urban areas with tall buildings, more than 3 scatterers is not that
  uncommon. (However, the claim of sparsity still holds well, thus this is just
  minor nitpicking on the way the claim is formulated.)

* Page 5, Figure 2: Is this figure necessary to include? In fact, the entire
  discussion section 3.2 seems unneccessary given the topic of the
  manuscript. Only the very high resolution is important, and this can be described
  with 1-2 sentences instead of a whole section and a figure.

* Page 5, Figure 3: The figure is too small, and the small images in the boxes
  on top and left are unreadable when printed on a normal office printer. Even
  on a high-contrast computer screen, it is not easy to read. Either make
  illustrative cartoons (preferred), or improve the quality of the images,
  e.g. by changing black background to white by inverting the colorscale.

* Page 5, line 205: The first sentence of Section 3.1 is a little unclear. It is
  unclear whether the concept of "joint sparsity" an original contribution of
  this manuscript. Probably not, and in that case, a general citation or two
  would be good at this point.

* Page 6, line 222: "Equation (6) can also be converted into [...]" -> this is
  not technically correct. Equation (7) is a formulation of a possible estimator
  for the unknown \Upsilon in (6), not a "converted equation".

* Page 7, line 236-237: "sub-millimeter accuracy" -> it is not possible to
  measure points with such accuracy. It is also not needed; Accuracy on the
  order of 1/4 pixel should be more than sufficent.

Author Response

Dear professor,

Thank you for your review. Please see the attached PDF file.

Best regards!

Lei Pang; Yanfeng Gai; Tian Zhang.

2020/10/03

Reviewer 2 Report

Please, see the attached PDF file.

Author Response

(The authors gave the same response as above.)

Round 2

Reviewer 1 Report

I recommend publishing the revised manuscript, subject to minor modifications.

  • The English must be improved.
  • In Equation (7), the regularisation term is not discussed: what was the choice of weight \lambda_\kappa, and was 1-norm or 2-norm used?
  • In my previous reply, I recommended a short discussion on the complexity of the problem. Run time has now been included, which gives an indication. However, I was more interested in the algorithmic complexity: is the solution of eq (7) NP-hard or not under reasonable boundary conditions? The runtime difference of 40x wrt an external software package might be due to the implementation quality rather than a real difference in algorithmic complexity.

Author Response

Dear professor,

        Thank you very much for your review and recognition of this manuscript. After the second English editing service of MDPI, the English grammar of this article has been well improved.  Please see the attached PDF file for the modifications of this manuscript.

Best regards!

Lei Pang; Yanfeng Gai; Tian Zhang.

2021/10/06

Reviewer 2 Report

Please see the attached pdf file

Author Response

Dear professor,

        Thank you very much for your review and recognition of this manuscript. After the second English editing service of MDPI, the English grammar of this article has been well improved.  

Thank you again for your recent help.

Best regards!

Lei Pang; Yanfeng Gai; Tian Zhang.

2021/10/06